# Help Seeking of Highly Specialized Mental Health Treatment before and during the COVID-19 Pandemic among Health Professionals

**DOI:** 10.3390/ijerph19063665

**Published:** 2022-03-19

**Authors:** María Dolores Braquehais, Esperanza L. Gómez-Duran, Gemma Nieva, Sergi Valero, Josep Antoni Ramos-Quiroga, Eugeni Bruguera

**Affiliations:** 1Integral Care Program for Sick Health Professionals, Galatea Clinic, Galatea Foundation, 08017 Barcelona, Spain; elgomezduran@comb.cat (E.L.G.-D.); gnieva@vhebron.net (G.N.); svalero@fundacioace.org (S.V.); ebruguer@vhebron.net (E.B.); 2Psychiatry, Mental Health and Addiction Research Group, Networking Research on Mental Health (CIBERSAM), Vall d’Hebron Institut de Recerca (VHIR), Vall d’Hebron Hospital Universitari, Vall d’Hebron Barcelona Hospital Campus, 08035 Barcelona, Spain; jaramos@vhebron.net; 3School of Medicine, Universitat Internacional de Catalunya, 08017 Barcelona, Spain; 4Department of Psychiatry, Vall d’Hebron Hospital Universitari, Vall d’Hebron Barcelona Hospital Campus, 08035 Barcelona, Spain; 5ACE Alzheimer Center Barcelona, Universitat Internacional de Catalunya (UIC), 08017 Barcelona, Spain; 6Department of Psychiatry, School of Medicine, Universitat Autònoma de Barcelona, 08193 Barcelona, Spain

**Keywords:** COVID-19, health professionals, mental health, treatment services, mental health programs, mental disorders, substance use disorders

## Abstract

(1) Background: Ongoing specialized programs for health professionals (HPs) adapted their treatment services during the COVID-19 pandemic. (2) Methods: We conducted a retrospective observational study of medical e-records of HPs with mental disorders working in Catalonia that were consecutively admitted to the Galatea Care Program Clinical Unit. The sample (N *=* 1461) was divided into two periods: 21.5 months before (*n* = 637) and after (*n* = 824) 14 March 2020. (3) Results: There was a significant increase (29.4%) in the number of referrals to the specialized Clinical Unit during the pandemic, especially with respect to physicians compared to nurses. The percentage of HP women at admission and the clinical severity of the first treatment episode remained without changes before and after the COVID-19 pandemic. The most prevalent main diagnoses also remained similar: adjustment disorders (41.5%), mood disorders (24.9%), anxiety disorders (14.4%), and substance use disorders (11.8%). (4) Conclusions: HPs, particularly physicians, more frequently sought voluntary help from specialized mental health programs during the COVID-19 pandemic. Future studies are needed to analyze the reasons behind this finding and the evolution of referrals to these types of programs after the COVID-19 outbreak.

## 1. Introduction

The novel coronavirus disease (COVID-19) pandemic caused a devastating global health crisis, beginning in 2020, infecting over 270 million people, and causing over 5 million deaths worldwide [1]. The World Health Organization also estimates that between 80,000 and 180,000 health professionals (HPs) could have died from COVID-19 in the period between January 2020 to May 2021, with a medium case scenario of 115,500 deaths [2]. In Spain, 6,922,466 cases had been confirmed and 89,837 had died by 5 January 2022 [3]. With respect to HPs, before 11 May 2020, 40,921 had been infected, 4177 needed hospitalization, and 53 died [4].

In many countries, healthcare systems have been overwhelmed with the serious depletion of basic resources, especially during the first wave of the pandemic. Besides work overload, HPs, especially those at the frontline of care, have been confronted with unexpected, more intense and frequent traumatic experiences than the general population [5]. Their most common sources of distress have been related to fear of contagion (both in themselves or in relatives), lack of protective measures, social stigma associated with COVID-19 exposure, ethical dilemmas, information and training, and aspects concerning perceived support by families, colleagues, institutions, and society [6,7].

The negative impact of the pandemic on the mental health of professionals during the pandemic has also been stressed by other authors [8], and feelings of being inadequately supported in this scenario may have contributed to impaired mental health [9,10]. The impact of these highly stressful circumstances and the lack of appropriate environmental resources to deal with them may have contributed to increased rates of burnout [11,12] and highlighted the importance of providing mental health support for those in trouble [10].

In a recent meta-analysis of 65 studies and 79,437 HPs worldwide, the average prevalence of anxiety, depression, stress, post-traumatic stress syndrome, insomnia, psychological distress, and burnout were 34.4%, 31.8%, 40.3%, 11.4%, 27.8%, 46.1% and 37.4%, respectively [13]. Estimated prevalence figures vary depending on epidemiological variables, such as number of cases per 100,000 habitants, specific COVID-19 pandemic stage, health service systems characteristics and vaccination rates. Regrettably, information regarding maladaptive coping strategies, such as alcohol use or sedative self-prescription, is less available [14,15]. Most studies do not specifically screen for potential substance use disorders, although experience from previous pandemics points to an increase in the incidence of addictive behaviors among HPs both in the short- and mid-to-long-term [16].

Women, nurses and frontline responders have more frequently developed anxiety and depression compared to men, doctors and second-line personnel [17,18,19]. In some studies, younger and less experienced HPs have also been reported to be at higher risk, [18] while resilience, perceived intimate and public support, and positive coping styles have been identified as protective factors [20]. The majority of studies are cross-sectional, report information on the beginning of the pandemic (first 3–6 months after each country’s epidemic outbreak) and many of them have been conducted in China [5,6,13]. Most of them describe the impact of the pandemic on HPs’ mental health, but little is known with respect to the effectiveness of mental health interventions for HPs during that period [21,22,23,24].

Physician health programs in the United States have adapted their provision of services and protocols both to support doctors and to continue monitoring those with substance use disorders in order to warrant safe practice [25]. In the United Kingdom, in the National Health Service (NHS) Practitioner Health Programme, nearly as many patients presented in the 12-month pandemic period (April 2020–March 2021) as in the first ten years of service (4355 in the last 12 months vs. 5000 over the first 10 years). Women far outweighed the number of men (78% vs. 22%), including when compared to the pre-pandemic distribution (67.5% women). Mean age at admission decreased from the pre-pandemic level. Some specialties increased both in relative and absolute frequencies (hospitalists, accident and emergency doctors, and pediatricians), others remained stable (psychiatry, diagnostics, and surgery) and some presented a percentage decrease (intensive care unit specialists and anesthesiologists). General practitioners saw the largest percentage decline as a proportion of the caseload, but remained disproportionally higher than their specialty size [26].

This study aims to compare the profiles of HPs with mental disorders accessing the Galatea Care Program Clinical Unit the year before and after the COVID-19 pandemic outbreak. HPs asking for help at the program but only needing brief counseling were excluded. Comparisons between main HP groups (physicians, nurses, and psychologists) were also conducted. Based on the relevant research evidence, we expected to find an increase in both the number of admissions and the number of nurses asking for help, and a decline in the mean age at admission compared to referrals before the lockdown. The prevalence of anxiety and mood disorders was also expected to have increased during the COVID-19 pandemic.

## 2. Materials and Methods

### 2.1. Design

This is an observational retrospective chart review study. Data from all e-medical records of HPs consecutively admitted to the Galatea Care Program were downloaded from the program databases in January 2022.

### 2.2. Participants

The sample was divided into two periods: 21.5 months before (*n* = 637) and 21.5 months after (*n* = 824) the official lockdown in Spain (14 March 2020). Therefore, a total of 1461 medical e-records of HPs working in Catalonia and admitted to the Galatea Care Program Clinical Unit from June 2018 until December 2021 were reviewed. 

### 2.3. Program Description

The Galatea Care Program is a free, voluntary, highly confidential, specialized mental health service run by the Galatea Foundation that offers prevention and treatment interventions for HPs with mental disorders, including addictions. In 1998, the physician health program (PAIMM in Catalan) was launched, in 2000 it was extended to nurses (RETORN program), and during the last decade it has also been offered to other HPs. The program is funded by both the Department of Health of the Catalan Government (*Generalitat de Catalunya*) and the HPs’ various association councils. All HPs benefiting from the program needed to be registered at their respective professional association councils. Registration is mandatory in Spain for HPs to be able to practice. 

### 2.4. Type of Intervention

HPs asking for help at the Galatea Care Program are briefly evaluated by a Galatea Foundation health care expert during a preliminary telephone interview. If only counseling is needed, they are offered up to 5 sessions of emotional support. However, if they need a thorough psychiatric or psychological assessment, they are referred to the program’s Clinical Unit, the Galatea Clinic, where they are offered outpatient, day hospital, or inpatient services depending on each case’s severity. The majority of patients only need outpatient treatment. Treatment in the program becomes mandatory to retain the license to practice only when risk and/or evidence of practice difficulties are identified. HPs accept this clause, among others related to service provision conditions, when they sign the informed consent form before entering the program. Nevertheless, HPs with risk of practice difficulties represent less than 5% of all HPs admitted to the program.

HPs are initially evaluated at the outpatient service by a psychiatrist or a psychologist who makes a diagnostic evaluation during the three first interviews, and reaches an agreement with the patient about his/her treatment plan. The Galatea Clinic inter-disciplinary approach covers the following intervention areas depending on each patient’s treatment plan: psychiatry, psychology, neuropsychology, social work, general health, and physiotherapy. Psychotherapy based on a combination of cognitive-behavioral and motivational therapy is delivered in individual or group format. 

During the pandemic, most individual interventions were conducted telematically (by phone or videoconference) although severe cases were preferably evaluated in person. When the COVID-19 incidence descended, individual consultations were again held in person. Outpatient group psychotherapy was always conducted by videoconference since the COVID-19 outbreak. During the pandemic, the inpatient unit continued its activity under specific infection prevention protocols. Some new online intensive programs were also developed to reach severe cases with limitations on being hospitalized. Drug screening was changed from weekly to random frequency. Patients with practice problems, higher suicidal risk and those who were known to be in the front line of care were more frequently followed. Supervision at the workplace was also intensified when necessary.

### 2.5. Demographic and Clinical Variables

Demographic (sex, age) and clinical (main diagnosis, hospitalization, type of referral) variables were obtained from each medical record. Type of referral to the program was divided into:Self-referrals;Directed referrals, including:Induced referrals: voluntary referrals encouraged or induced by managers, colleagues or relatives;Referrals after confidential information or a formal malpractice claim was received by the Barcelona Nurse Association Council.

Main diagnoses followed DSM-IV-TR criteria [27]. Need of hospitalization during the first treatment episode was considered an indicator of clinical severity.

### 2.6. Statistical Analyses

Besides descriptive analysis, chi-square tests were used to compare dichotomous variables between the two selected groups (HPs admitted before and after lockdown). ANOVA test was used to compare quantitative variables over time and with Chi-square test to compare qualitative variables. Main diagnoses were grouped into substance use disorders, mood disorders (including bipolar disorders), adjustment disorders, anxiety disorders (including obsessive compulsive disorder), and others. Odds ratios with 95% confidence intervals were used to analyze the relationship between binary variables. 

A multivariate logistic regression analysis differentiating HPs accessing the program and needing psychiatric assessment before and after 14 March 2020 was executed with the variables identified to be statistically different in the preliminary bivariate analysis. All statistical analyses were executed for the global sample and for the main HP groups (physicians, nurses, and psychologists). The hypothesis tests were two-tailed and conducted with an alpha of 0.05. All statistical analyses were conducted using STATA v.15 (StataCorp. 2017. *Stata Statistical Software: Release 15*. StataCorp LLC, College Station, TX, USA).

## 3. Results

### 3.1. All Health Professionals (HPs)

Of all the HPs accessing the Galatea Clinical Unit during that period (*n* = 1461), the majority were women (*n* = 1064; 72.8%), their mean age was 46.3 (SD = 11.7) years, and they were predominantly self-referred (*n* = 1394; 97%). Most of them were physicians (*n* = 905; 61.9%), followed by nurses (*n* = 403; 27.6%), psychologists (*n* = 52; 3.6%), social workers (*n* = 37; 2.5%), vets (*n* = 28; 1.9%), dentists (*n* = 18; 1.2%), and pharmacists (*n* = 18; 1.2%). The highest increase in referrals during the pandemic was observed among physicians (see Table 1). 

Overall, the most prevalent diagnoses at admission were adjustment disorders (*n* = 606; 41.5%), followed by mood disorders (*n* = 364; 24.9%), anxiety disorders (*n* = 210; 14.4%), substance use disorders (*n* = 172; 11.8%), and others (*n* = 109; 7.5%). Regarding hospitalization during the first treatment episode, almost a sixth of the whole sample were admitted to the inpatient unit (*n* = 206; 14.8%).

During the COVID-19 pandemic, the number of admissions to the Galatea Care Program Clinical Unit increased by 29.4% compared to the previous period (*n* = 824 vs. *n* = 637). Differences for HPs admitted before and after official lockdown in Spain (14 March 2020) are summarized in Table 2.

Differences for physicians, nurses, and psychologists are reported in Table 3.

### 3.2. Physicians

Physicians admitted to the Galatea Care Program Clinical Unit during the whole period (*n* = 905) were more frequently women (*n* = 596; 65.9%), their mean age was 46 (SD = 12.2) years, and the vast majority were self-referred (*n* = 862; 97.1%). Main diagnoses were adjustment disorders (*n* = 400; 44.2%), mood disorders (*n* = 230; 25.4%), anxiety disorders (*n* = 119; 13.1%), substance use disorders (*n* = 92; 10.2%), and others (*n* = 64; 7.1%). One out of ten needed hospitalization (*n* = 116; 12.8%).

Physicians admitted during the pandemic were more likely to be self-referred (χ^2^ = 8.2; df = 1; *p* < 0.01; OR = 0.29 (95%CI: 0.12–0.67)), and less frequently needed hospitalization during the first treatment episode (χ^2^ = 5.9; df = 1; *p* = 0.02; OR = 0.61 (95%CI: 0.41–0.9)) compared to those admitted before the official lockdown. No differences were found between those admitted before and after the pandemic, regarding sex, mean age, or main diagnoses.

### 3.3. Nurses

Nurses referred to the Galatea Care Program Clinical Unit during the whole period (*n* = 403) were predominantly women (*n* = 351; 87.1%), their mean age was 47.3 (SD = 10.9) years, and they were more frequently self-referred (*n* = 383; 96.5%). The main diagnoses were adjustment disorders (*n* = 147; 36.5%), mood disorders (*n* = 99; 24.6%), anxiety disorders (*n* = 70; 17.4%), substance use disorders (*n* = 54; 13.4%), and others (*n* = 33; 8.2%). Almost one fifth needed hospitalization (*n* = 70; 17.4%).

No statistically significant differences were found between nurses admitted before (*n* = 213) and after (*n* = 190) the lockdown, with respect to sociodemographic information, type of referral, main diagnosis, and clinical severity.

### 3.4. Psychologists

Globally, psychologists (*n* = 52), admitted to the Galatea Clinical Unit during the whole period were also more frequently women (*n* = 43; 82.7%), their mean age was 41.5 (SD = 10.3) years, and most of them were self-referred (*n* = 48; 94.1%). The most prevalent main diagnoses at admission were adjustment disorders (*n* = 18; 34.6%), followed by mood disorders (*n* = 11; 21.2%), substance use disorders (*n* = 10, 19.2%), anxiety disorders (*n* = 6; 11.5%), and others (*n* = 7, 13.5%). Almost a fourth needed hospitalization (*n* = 14, 26.9%).

No differences were found between psychologists admitted before (*n* = 22) and after (*n* = 30) the lockdown.

## 4. Discussion

The pandemic is known to have increased the mental distress not only of the general population, but also of people at the frontline of care, including HPs and others exposed to some degree of vicarious trauma [8,9,10,11,12,28,29]. Several temporary initiatives have been developed worldwide to help HPs cope with high levels of distress [30]. However, this is the first study that has specifically analyzed the trends in admissions to a well-established, highly specialized mental health unit for HPs (not merely physicians), before and after the COVID-19 outbreak. The total time span of this study covers ~39 months before and after the pandemic outbreak. Therefore, it provides a wider perspective than epidemiological studies conducted on HPs during the COVID-19 crisis, as they were mostly conducted during the initial phases of the pandemic. Another novelty of this study compared to other intervention studies is that it is gives information about a specialized mental health unit for HPs. Moreover, while the NHS specialized program only offers treatment for physicians and dentists in the UK [26], the Galatea Care Program in Catalonia is open to other healthcare professionals. 

The number of referrals to the Galatea Clinical Unit significantly increased during the pandemic period, especially among physicians. Although they are known to resist voluntarily seeking treatment help [31], the frequency of physicians’ referrals increased during that period. In fact, they were more likely to be self-referred and less frequently needed hospitalization during the first treatment episode (an indirect indicator of clinical severity). This may show their confidence in a well-known program that has been running for more than two decades in Catalonia. It could also be attributed to the broad societal sympathy and appreciation that may have lessened perceived stigma, as well as to the serious consequences of the pandemic on their mental health that exceeded the professionals’ threshold of tolerance to discomfort, to the point of overcoming their difficulties in asking for help. On the other hand, the other HPs’ profiles, with respect to mean age at admission, percentage of women, main diagnosis prevalence, and clinical severity did not significantly differ before and after the official lockdown. 

In our sample, the percentage of women seeking help remained unchanged when comparing admissions before and after the pandemic. This is different from data reported from the UK’s NHS program for physicians and dentists [26], and from the scientific evidence on the higher emotional impact of the pandemic on female HPs [17,18,19]. This discrepancy could be related to the fact that this is the first program for physicians in Europe, so gender trends may have been stable well before the pandemic outbreak. In fact, the percentage of women seeking help during the pandemic from the UK’s NHS program was 75% (compared to the 65% previously) [21], while in our program it remained around 65% among physicians and 72% among other HPs.

The age at admission was also similar in both groups, although some studies point to a higher risk of developing mental distress among younger and less experienced HPs [18]. Reasons behind this finding should be carefully considered, as it may be related to more difficulties in self-recognition of mental distress in this group and to resistances to seek appropriate treatment help. However, this does not comply with the increase in younger physicians’ referrals to the Galatea program observed in 2020 [32,33].

An increase in the number of nurses admitted to the program was also expected to occur [17,18,19]. However, we found that the most significant raise happened among physicians, and the proportion of main diagnosis at admission among physicians and nurses were different. Although mood, anxiety, and adjustment disorders accounted for the highest percentage of diagnosis in all HPs profiles, the proportion of substance use disorders decreased among physicians and increased among nurses and psychologists. This can be associated with several factors. Mental health-related stigma has been described both in physicians and nurses, but reluctance to seek help may be greater in physicians even as early as the undergraduate stage [26]. Nurses may be more prone to seek help at their workplace, where numerous mental health support initiatives are available. They may also be less fearful about recognizing mental distress and asking for help when in trouble, thus probably benefiting from early counseling interventions that prevent mental disorders needing highly specialized mental health treatment. In contrast, physicians may delay help seeking what increases the risk of mental distress turning into a mental disorder. In that situation, they feel more comfortable when being in treatment at a highly confidential program where their identity is concealed.

Data on the impact of the COVID-19 pandemic among HPs has shown an increase in the incidence of mental disorders among them, mainly of anxiety and mood disorders [13]. This was also the case in our sample. In fact, adjustment disorders are considered a subtype of anxiety disorders related to recent stressful life events. Although there was also an increase in the number of referrals for substance use disorders during the pandemic, this did not parallel the greater increase in admissions related to mood and anxiety disorders. Despite of the fact that there is less scientific evidence on the impact of the pandemic on alcohol and self-prescribed drug misuse among HPs [14,15], our findings point to the need to adequately prevent and screen for this maladaptive coping strategy among HPs, especially among physicians, confronted by such highly challenging circumstances.

The main strengths of this study were: (1) the information was obtained from a highly specialized mental health program for HPs (not only physicians and nurses) already running when the pandemic began; (2) the sample size; and, (3) the large time span, including a pre-COVID period. Its main weaknesses were: (a) its design, as it was a retrospective chart review; (b) the diagnoses were not obtained after a structured interview; (c) the lack of data in terms of personality traits and/or other psychosocial aspects that could enrich the comprehension of the similarities and differences between groups; (d) the difficulties in generalizing findings to other social and cultural environments; and, (e) the need for information on some variables associated with COVID-19 in the pandemic subsample. Our findings should also be interpreted cautiously as the specific procedures and characteristics of our program need to be considered when generalizing our conclusions to other treatment settings [27].

## 5. Conclusions

COVID-19 had and continues to have a negative impact on the mental health of the general population, but some groups, such as HPs or others exposed to vicarious trauma, have endured even more stressful circumstances. Highly specialized mental health programs for HPs have experienced an increase in the number of referrals during the pandemic. Despite their known resistance to seeking help when suffering from mental disorders, physicians have felt sufficiently confident to self-refer to a specialized mental health unit during the pandemic. Reaching younger HPs in these circumstances should be a priority, as they are known to have suffered from more mental distress during the pandemic. Adjustment, mood, and anxiety disorders remain the most frequent diagnoses. Substance use disorders still represent one in ten of all referrals, although the absolute frequency of referrals did not significantly increase during the pandemic. Admissions to programs for HPs are likely to remain higher than before COVID-19, according to the research evidence on the impact of the pandemic on their wellbeing, and once the temporary mental health interventions developed during the pandemic will no longer be available. Further analyses will provide more information on admission trends to these specific programs and on factors related to the onset of mental disorders among HPs under these stressful circumstances. Future research lines could also analyze the efficacy of this type of highly specialized intervention provided to health professionals during the pandemic. Finally, in addition to offering mental health support, providing HPs with adequate protective measures and sufficient human and material resources during epidemic crises should be a priority, as this can contribute to reducing mental distress associated with uncertainty and work overload. Caring for those who care for others should become a main concern not only for healthcare institutions but also for society as a whole.

## Figures and Tables

**Table 1 ijerph-19-03665-t001:** Health professionals admitted to the Galatea Care Program Clinical Unit before and after the COVID-19 official lockdown (*n* = 1461).

Occupation	Before Lockdown (*n* = 637)	After Lockdown (*n* = 824)
Physicians, *% (*n*)*	57 (363)	65.8 (542)
Nurses, *% (*n*)*	33.4 (213)	23.1 (190)
Psychologists, *% (*n*)*	3.5 (22)	3.6 (30)
Social workers, *% (*n*)*	2.5 (16)	2.5 (21)
Vets, *% (*n*)*	1.3 (8)	2.4 (20)
Pharmacists, *% (*n*)*	1.3 (8)	1.2 (10)
Dentists, *% (*n*)*	1.1 (7)	1.3 (11)

**Table 2 ijerph-19-03665-t002:** Sociodemographic, occupational and clinical characteristics of HPs admitted to the Galatea Care Program Clinical Unit before and after the COVID-19 official lockdown *n* = 1461).

Variables	Before Lockdown (*n* = 637)	After Lockdown (*n* = 824)	Statistics		
			t/χ^2^	*p*	OR (95% CI)
Demographics					
Age, mean (SD)	44.11 (11.92)	45.06 (11.54)	1.54	0.13	0.08 ^1^
Women, % (*n*)	72.8 (464)	72.8 (600)	<0.01	1	1 (0.79–1.26)
Main mental health disorder					
Mood disorders, % (*n*)	24 (153)	25.6 (211)	3.31	0.51	
Anxiety disorders, % (*n*)	14.8 (94)	14.1 (116)			
Adjustment disorders, % (*n*)	40 (255)	42.6 (351)			
SUD disorders, % (*n*)	13 (83)	10.8 (89)			
Other disorders % (*n*)	8.2 (52)	6.9 (57)			
Type of referral					
Self-referrals ^2^, % (*n*)	95.8 (600)	97.9 (794)	4.47	0.04	0.49 (0.27–0.92)
Clinical severity					
Hospitalization % (*n*)	16.8 (107)	13.2 (109)	3.36	0.07	0.76 (0.57–1.01)

^1^ For quantitative variables, Cohen’s *D* are calculated; ^2^
*n* = 1437.

**Table 3 ijerph-19-03665-t003:** Sociodemographic, occupational and clinical characteristics of physicians (n = 905), nurses (n = 403), and psychologists (n = 52) admitted to the Galatea Care Program Clinical Unit before and after the COVID-19 official lockdown.

Variables	Before Lockdown	After Lockdown	Statistics		
			t/χ^2^	*p*	OR (95% CI)
**Physicians**					
Demographics					
Age, mean (SD)	43.9 (12.59)	44.9 (11.98)	1.2	0.23	0.08 ^1^
Women, % (*n*)	62.5 (227)	68.1 (369)	2.73	0.1	1.28 (0.97–1.69)
Main mental health disorder					
Mood disorders, % (*n*)	25.9 (94)	25.1 (136)	4.75	0.31	
Anxiety disorders, % (*n*)	13.8 (50)	12.7 (69)			
Adjustment disorders, % (*n*)	40.5 (147)	46.7 (253)			
SUD disorders, % (*n*)	12.1 (44)	8.9 (48)			
Other disorders, % (*n*)	7.7 (28)	6.6 (36)			
Type of referral					
Self-referrals ^2^, % (*n*)	95 (339)	98.5 (523)	8.19	<0.01	0.29 (0.12–0.67)
Clinical severity					
Hospitalization % (*n*)	16.3 (59)	10.5 (57)	5.9	0.02	0.61 (0.41–0.9)
**Nurses**					
Demographics					
Age, mean (SD)	44.82 (11.1)	46.2 (10.8)	1.22	0.22	0.12 ^1^
Women, % (*n*)	89.2 (190)	84.7 (161)	1.41	0.24	0.67 (0.37–1.21)
Main mental health disorder					
Mood disorders, % (*n*)	21.1 (45)	28.4 (54)	4.81	0.31	
Anxiety disorders, % (*n*)	17.4 (37)	17.4 (33)			
Adjustment disorders, % (*n*)	40.8 (87)	31.6 (60)			
SUD disorders, % (*n*)	13.1 (28)	13.7 (26)			
Other disorders % (*n*)	7.5 (16)	8.9 (17)			
Type of referral					
Self-referrals ^3^, % (*n*)	96.7 (202)	96.3 (181)	>0.01	1	1.12 (0.38–3.24)
Clinical severity					
Hospitalization % (*n*)	15.5 (33)	19.5 (37)	0.85	0.36	1.32 (0.79–2.21)
**Psychologists**					
Demographics					
Age, mean (SD)	39.09 (10.09)	40.53 (10.37)	0.5	0.62	0.14 ^1^
Women, % (*n*)	77.3 (17)	86.7 (26)	0.26	0.61	1.91 (0.45–8.15)
Main mental health disorder					
Mood disorders, % (*n*)	18.2 (4)	23.3 (7)	6.47	0.17	
Anxiety disorders, % (*n*)	9.1 (2)	13.3 (4)			
Adjustment disorders, % (*n*)	31.8 (7)	36.7 (11)			
SUD disorders, % (*n*)	13.6 (3)	23.3 (7)			
Other disorders % (*n*)	27.3 (6)	3.3 (1)			
Type of referral					
Self-referrals ^4^, % (*n*)	95.2 (20)	93.3 (28)	>0.01	1	1.43 (0.12–16.86)
Clinical severity					
Hospitalization % (*n*)	22.7 (5)	30 (9)	0.07	0.79	1.46 (0.41–5.17)

^1^ For quantitative variables, Cohen’s D are calculated; ^2^ n = 888; ^3^ n = 397; ^4^ n = 51.

## Data Availability

Due to the highly confidential nature of the Galatea Care Program for Health Professionals, no data set is provided.

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
