# Peer review of "Help Seeking of Highly Specialized Mental Health Treatment before and during the COVID-19 Pandemic among Health Professionals"

_ijerph, 2022, doi:10.3390/ijerph19063665_

Round 1
Reviewer 1 Report
The authors Braquehais et al explored help seeking of highly specialized mental health treatment before and during the COVID pandemic amongst health professionals. Even though the topic is interesting and highly relevant considering the continuation of the COVID pandemic and the consequences on health care professionals, some minor and major revisions have to be made in order to be accepted for publication.
Minor:
-refer to the total sample as N and refer to sub samples (for instance women in line 169 with n)
-table 2; do not display p in capital letter
-display all statistics in italic
Major:
-Can health care personal benefit from their findings? Discuss in general. The mental health of professionals during the pandemic is also stressed by other authors (Sandesh et al., 2020) and feelings of being inadequately supported may contribute to impaired mental health (Chen et al., 2020; de Vroege & van den Broek, 2021), please describe their own findings with regards to the other findings and implication for health care professionals. Furthermore, a lot of (Chinese but also other countries) literature is available on this topic. Burnout can be induced (Li et al., 2020; Ornell and Schuch, 2020) and the call for support of mental care is high (de Vroege & van den Broek, 2021). Please link these findings to your findings in an extra paragraph in the manuscript to generalize the findings to a broader public that suffers from covid pandemic as well.
References suggestions to be added:
de Vroege, L., and van den Broek, A. (2021). Results of mental support for health care professionals and mental care during the COVID-19 pandemic. J Public Health (Oxf). doi: 10.1093/pubmed/fdaa278.
World Health Organization (2020). Mental health and psychosocial considerations during the covid-19 outbreak [Online]. Available: https://www.who/int/docs/default-source/coronaviruse/mental-health-considerations.pdf
Sandesh, R., Shahid, W., Dev, K., Mandhan, N., Shankar, P., Shaikh, A., et al. (2020). Impact of COVID-19 on the Mental Health of Healthcare Professionals in Pakistan. Cureus 12(7), e8974. doi: 10.7759/cureus.8974.
Chen, Q., Liang, M., Li, Y., Guo, J., Fei, D., Wang, L., et al. (2020). Mental health care for medical staff in China during the COVID-19 outbreak. Lancet Psychiatry 7(4), e15-e16. doi: 10.1016/s2215-0366(20)30078-x.
Ornell, F., and Schuch, J.B. (2020). "Pandemic fear" and COVID-19: mental health burden and strategies. 42(3), 232-235. doi: 10.1590/1516-4446-2020-0008.
de Vroege L & van den broek A (2021). Substantial impact of COVID-19 on self-reported mental health of healthcare professionals in the Netherlands. Frontiers in public health, epub available at https://www.frontiersin.org/articles/10.3389/fpubh.2021.796591/abstract
Li, Z., Ge, J., Yang, M., Feng, J., Qiao, M., Jiang, R., et al. (2020). Vicarious traumatization in the general public, members, and non-members of medical teams aiding in COVID-19 control. Brain Behav Immun 88, 916-919. doi: 10.1016/j.bbi.2020.03.007.
Author Response
We thank the reviewer for his/her comments. The original and reviewed version of the manuscript have been proofread by a native English speaking editor.
We have unified the format to refer to the total sample (N) and subsamples (n) size. We have also corrected the “p” (it was wrongly corrected by the Word program in capital letters) and have displayed the statistics in italics.
We have expanded the Introduction (see lines 51-57, 74) and Discussion (see lines 242-244; 307-310) to include some of these comments. We have also included the references suggested by the reviewer. We have added two comments to relate our findings to the general public in the Conclusion section (see lines 320-322 and lines 340-342).
Reviewer 2 Report
The authors performed an interesting retrospective observational study of e-medical documentation regarding different health professionals treated for mental disorders. The sample was divided into pre and after declared covid epidemic.
Strengths: -possibility to monitor of highly specialized mental health programs specific for health professionals (stigma problem);- large sample; -subgroup of health professionals; lack of studies on large samples;
Limitations:- results not applicable to all health professionals with mental disorder, especially addiction; - results applicable to special setting and social and cultural environment; - lack of clinical assessment (e- medical documentation; - diagnoses were not obtained after a structured interview;
I miss some strengths of the study
Author Response
We thank the reviewer for his/her positive feedback and have expanded the discussion with respect to the strengths and limitations of the study according to the reviewer’s comments (see lines 307-318).
Reviewer 3 Report
The authors retrospectively conducted an observational study of medical e-records of health professionals with mental disorders working in Catalonia. The paper is well-written, properly justified, and potentially impactful. I have no major concerns about this paper and I would recommend its publications in the present form.
The only additional thing I can suggest is to foresee in future works further evidence about the efficacy of the intervention provided to health professionals during the pandemic.
Author Response
We thank the reviewer for his/her feedback. We have added a comment on lines of research that could be explored in the future (see lines 336-337).
Reviewer 4 Report
It is a paper totally adjusted to the journal and to the Mental Health monograph, it is written in correct scientific language and satisfies the formal requirements of the journal and of science.
Author Response
We thank the reviewer for his/her positive feedback.